# Light-Regulated Transcription of a Mitochondrial-Targeted K^+^ Channel

**DOI:** 10.3390/cells9112507

**Published:** 2020-11-19

**Authors:** Anja J. Engel, Laura-Marie Winterstein, Marina Kithil, Markus Langhans, Anna Moroni, Gerhard Thiel

**Affiliations:** 1Membrane Biophysics and Center for Synthetic Biology, Technische Universität Darmstadt, 64287 Darmstadt, Germany; anja.jeannine.engel@gmx.de (A.J.E.); Laura.schlee@aol.com (L.-M.W.); mekithil@gmail.com (M.K.); 2Ernst-Berl-Institute for Technical and Macromolecular Chemistry, Technische Universität Darmstadt, 64287 Darmstadt, Germany; Langhans@bio.tu-darmstadt.de; 3Department of Biosciences and CNR IBF-Mi, Università degli Studi di Milano, 20133 Milano, Italy; Anna.moroni@unimi.it

**Keywords:** optogenetic platform, mitochondrial K^+^ channel, mitochondrial Ca^2+^ manipulation, apoptosis

## Abstract

The inner membranes of mitochondria contain several types of K^+^ channels, which modulate the membrane potential of the organelle and contribute in this way to cytoprotection and the regulation of cell death. To better study the causal relationship between K^+^ channel activity and physiological changes, we developed an optogenetic platform for a light-triggered modulation of K^+^ conductance in mitochondria. By using the light-sensitive interaction between cryptochrome 2 and the regulatory protein CIB1, we can trigger the transcription of a small and highly selective K^+^ channel, which is in mammalian cells targeted into the inner membrane of mitochondria. After exposing cells to very low intensities (≤0.16 mW/mm^2^) of blue light, the channel protein is detectable as an accumulation of its green fluorescent protein (GFP) tag in the mitochondria less than 1 h after stimulation. This system allows for an in vivo monitoring of crucial physiological parameters of mitochondria, showing that the presence of an active K^+^ channel causes a substantial depolarization compatible with the effect of an uncoupler. Elevated K^+^ conductance also results in a decrease in the Ca^2+^ concentration in the mitochondria but has no impact on apoptosis.

## 1. Introduction

Ion channels comprise sophisticated membrane proteins for a selective and regulated passage of ions across an otherwise impermeable membrane barrier [1]. Given their central role in many physiological processes, they have been exploited as molecular tools to actuate ion fluxes and electrochemical potentials across membranes. In particular, major efforts have been undertaken to control ion channel function with light and thus regulate numerous cellular activities with high spatial and temporal precision such as neuronal firing yielding powerful tools for neurobiological research [2,3,4,5]. 

Current work extends these endeavors in finding ways to also regulate the activity of ion fluxes in the membranes of organelles. One interesting target is the inner membrane of mitochondria [6,7], which harbors several types of K^+^ channels [8,9,10]. A most recent review lists 10 distinct types of mitochondrial K^+^ channels with different conductance and gating features and discusses their structure/function properties and presumed physiological roles [11]. While a body of circumstantial evidence underpins an important role of these proteins in the regulation of mitochondrial membrane potential, calcium homeostasis, mitochondrial matrix volume and respiration [10,11,12], the precise function of K^+^ channels in mitochondria is not yet fully understood. More detailed information on K^+^ fluxes in mitochondria could be obtained from an optogenetic manipulation of potassium channels in these organelles. The power of such optogenetic experiments has already been recognized and successfully employed for a manipulation of mitochondrial processes by light. This includes a light-triggered manipulation of mitochondrial redistribution [13] interruption of mitochondria/ER contact sites [14] or an inhibition of mitophagy [15]. So far, only three optogenetic tools were reported for a manipulation of the mitochondrial membrane potential. The light-activated proton pump from *Leptosphaeria maculans* (Mac) was fused to the inner mitochondrial membrane protein (iMM) to generate mitochondria-ON [16]. This tool allows for a light-dependent increase in the mitochondrial membrane potential. It was reported that the light-triggered hyperpolarization of the mitochondria led to increased ATP production in *Caenorhabditis elegans* and that the activation of mitochondria-ON under hypoxic conditions prevented cytoprotection by mitochondrial preconditioning. Moreover, two channelrhodopsin-based tools were used for an optogenetic depolarization of the mitochondria. Tkatch and coworkers fused six repeats of the mitochondrial targeting sequence (MTS) from cytochrome C oxidase VIII (COX8) to the N-terminus of ChR2 to target the opsin to the iMM [6]. Blue-light illumination of the construct caused reversible depolarization of the mitochondria and reduced mitochondrial calcium uptake. The second mitochondria targeted channelrhodopsin was made by fusing the MTS of the ATP-binding cassette (ABC) transporter to ChR2 [7]; upon illumination, the mitochondria were depolarized. 

While experiments with ChR2 in mitochondria suggest a causal relationship between membrane voltage and physiological reactions, like the induction of apoptosis and a corruption of Ca^2+^ homeostasis, the data are not able to relate these effects to the activity of mitochondrial K^+^ channels. ChR2 is a non-selective cation channel and passes not only K^+^ but also Ca^2+^ and H^+^ [17]. Hence, the physiological consequences of ChR2 activation could originate from a membrane depolarization but also from the dissipation of Ca^2+^ and/or H^+^ gradients. In order to mimic the functional impact of endogenous K^+^ selective channels in the inner membrane, we employed a light-sensitive transcription system based on CRY2/CIB1 interaction [18]. This system is based on the blue-light activated plant cryptochrome 2 (CRY2) from *Arabidopsis thaliana* [18], which interacts after photoexcitation with partner proteins such as CIB1 [19]. The consequent heterodimerization of CRY2 and CIB1 has been engineered for an induction of transcription of various target genes with light [20]. By adopting this system for small K^+^ channels, we can trigger by light the synthesis of two small K^+^ channels, which are sorted into the inner membrane of mitochondria. After a lag time of ≤1 h it is possible to monitor the impact of an active K^+^ channel versus an inactive channel on the mitochondria. The data show that activation of the channel causes the expected depolarization of the mitochondrial membrane, a decrease in Ca^2+^ concentration in the organelles but no apoptosis.

## 2. Materials and Methods

### 2.1. Constructs

Light-sensitive transcription was achieved by co-expressing CIB1-VP64 (CIB1) and CRY2-Gal4BD (CRY2) together with channel protein of interest as described previously [20]. To optimize the system, the three essential genes were unified on one plasmid. Therefore, CIB1-VP64 and CRY2-Gal4BD were put under control of a cytomegalovirus CMV promotor (Appendix A). In this way, both proteins are transcribed together, before they are split into two independent proteins by a P2A self-cleaving site between the two proteins. The third component, the gene of interest (GOI) with the n-terminal 6xGalBS and the c-terminal of the green fluorescent protein (GFP) tag was cloned downstream of the CRY2-CIB1 coding region. The genes of interests are in the present study coding for K^+^ type channels protein. The Kesv protein (NCBI Accession #: NP_077708.1)) is coded by the Ectocarpus siliculosus virus 1 (EsV-1) [21] and Kmp_12T_ (YP_007676152) by Micromonas pusilla virus 12T [22].

### 2.2. Mutagenesis

All constructs for imaging were cloned into the standard peGFP-N2 vector (BD Biosciences Clontech, Heidelberg, Germany). Constructs used for planar lipid bilayer experiments were cloned into pET24-dellac (Merck, Darmstadt, Germany). Fragments were amplified with relevant overhangs using either Phusion polymerase (Thermo Fisher, Langenselbold, Germany) or Q5 polymerase (NEB, Frankfurt am Main, Germany) according to manufacturer’s specifications. The fragments were subsequently fused by either overlap-extension PCR [23] or Gibson Assembly [24]. The correct construct size was checked by standard agarose-gelelectrophoresis. Bands showing the expected size were purified with the Zymoclean Gel DNA Recovery Kit (Zymo Research, Freiburg, Germany) and the purified DNA was taken for heat shock transformation in competent *Escherichia coli* (DH5α). The amplified plasmid DNA was extracted with the ZR Plasmid Miniprep Kit Classic (Zymo Research, Freiburg, Germany) and sent sequenced (Microsynth Seqlab, Göttingen, Germany).

### 2.3. Heterologous Expression and Illumination Setup

Localization of the eGFP-tagged channels was performed in human embryonic kidney (HEK293) and Cos7 cells. All cell lines were cultured at 37 °C and 5% CO_2_ in T25 cell culture flasks in an incubator with DMEM/F12-Medium with Glutamine (Biochrom AG, Berlin, Deutschland) plus 10% fetal calf serum (FCS) und 1% Penicillin/Streptomycin. For imaging, cells were placed 48 h prior to examination on sterilized glass coverslips (No. 1.0; Karl Hecht GmbH & Co. KG, Sondheim, Germany) with a Ø = 25 mm. The cells were incubated for ~24 h at 37 °C with 5% CO_2_. As soon as the cells had reached a confluence of 60%, they were transfected with the appropriate plasmids. GeneJuice (Novagen, EMD Millipore Corp.; Billerica, MA, USA) or TurboFect^TM^ (Life Technologies GmbH; Darmstadt, Germany) were used as transfection reagents according to manufacturer specifications. Unless otherwise stated, 1 μg of plasmid DNA of the corresponding construct was always used or, in the case of co-transfection, 0.5 μg of each of the desired constructs.

For the illumination of cells, a custom-made illumination setup was used, which contained six 450 nm LEDs (Winger WEPRB3-S1 Power LED Star, 3W) which were arranged to fit under a standard 6-Well cell culture plate. For timing illumination protocols, the LEDs were attached to a timer that allowed for application of light pulses of defined length on a second to minute timescale. For the experiments shown here, pulsed blue light of 6 to 120 µmol of photons was applied for variable lengths of time. Overall incubation time between transfection and imaging was in all cases 16 h.

### 2.4. Confocal Microscopy

Confocal imaging was performed on either a Leica TCS SP or a Leica TCS SP5 II (Leica GmbH, Heidelberg, Germany). Cells seeded on coverslips were clamped into custom-made aluminum rings for imaging 16 h post transfection covered with 500 µL of phosphate buffer saline (PBS ) medium (8 g/L sodium chloride, 0.2 g/L potassium chloride, 1.42 g/L disodium hydrogen phosphate, 0.24 g/L potassium hydrogen phosphate; pH was adjusted with 1 M sodium hydroxide up to 7.4). Cells were imaged with either PL APO 100 × 1.40 OIL UV or HCX PL APO 63 × 1.20 W CORR UV objectives. Dyes or fluorescent proteins were excited with an argon (488 nm), krypton (568 nm) or helium-neon (632 nm) laser.

### 2.5. Planar Lipid Bilayer Experiments

The channel proteins for planar lipid bilayer were expressed in vitro into nanodiscs, purified and functionally reconstituted onto planar lipid bilayers as described previously [25]. 

### 2.6. Flow Cytometry

The relative polarization of the mitochondrial membrane was measured as described previously [26] using the accumulation of the fluorescent mitochondrial marker tetramethylrhodamine methyl ester perchlorate (TMRM) as an indirect measure of the mitochondrial membrane potential. HEK293 cells were measured >18 h after transfection with the Kesv channel plus GFP tag. After the removal of medium, cells were stained for 30 min with 10 nM TMRM in 1 mL PBS. After harvesting cells with Accutase, they were washed with fresh PBS and pelleted at 700 g for 5 min. The pellet was resuspended in 300 μL PBS and filtered with a 40 μm filter. The fluorescence was measured with S3e Cell Sorter (Bio-Rad GmbH; Munich, Germany). >10,000 cells per sample were analyzed by excitation with a 488 nm laser and by detection of TMRM fluorescence at 585 ± 42 nm. HEK293 cells treated for 5 min with the uncoupler carbonyl cyanide m-chlorophenylhydrazone (CCCP, 100 µmol) were used as a positive control. Data were analyzed with the FlowJo (FlowJo, LLC; Ashland, OR, USA) software.

## 3. Results and Discussion

To engineer a platform for the light-inducible transcription of a K^+^ channel for the inner membrane of mitochondria (iMM), we were choosing two small viral coded K^+^ channels (Appendix A), which are both sorted in mammalian cells into the mitochondria. Transient expression of both channels with a c-terminal GFP tag from genes, which were codon optimized for mammalian cells (Appendix A), results in HEK293 cells in an accumulation of GFP fluorescence in the mitochondria (Figure 1A–D). Colocalization with an organelle-specific marker underpins that Kesv is sorted into the mitochondria (Figure 1A,D). Moreover, the second channel, Kmpv_12T_, is in the majority of cells targeted to the mitochondria with no evidence for entry into the secretory pathway (Figure 1B,D). In a small population of cells, however, we also detected a high background of GFP fluorescence throughout the entire cell, including the nucleus (Figure 1C,D). The presence of GFP in the nucleus suggests that at least some of the channel protein must have been degraded, allowing GFP diffusion into the nucleus.

To obtain insight into the functional properties of the two channels, they were reconstituted in planar lipid bilayers. Frequent reconstitution attempts of the Kesv protein generated no detectable current in any of the recordings. Since viral K^+^ channels are generally easy to measure in planar lipid bilayers [25,27,28] the data suggest that the channel is not functional. To further test this assumption, Kesv was expressed in HEK293 cells and the membrane potential in the mitochondria monitored by flow cytometry. Appendix A shows the results of a typical measurement. The fluorescence intensity is higher in cells stained with TMRM than in unstained controls confirming the accumulation of the dye in the mitochondria. The fluorescence signal from cells expressing Kesv (purple) is undistinguishable from that of un-transfected HEK293 cells but much higher than in un-transfected cells treated with 100 µmol CCCP. The results of these experiments confirm that TMRM fluorescence reports the membrane polarization of mitochondria in cells and that an expression of Kesv has no perceivable effect on this parameter; this is consistent with the bilayer recordings in that the channel is not active. 

It had already been reported that Kmpv_12T_ generates in planar lipid bilayers robust K^+^ selective channel activity [22]. The same activity was found when the protein was expressed in vitro (Figure 2A). This protein generates in phosphatidylcholine bilayers a high unitary channel conductance of about 40 pS (Figure 2B). The channel has a low open probability, which increases in the setting of the bilayer at negative voltages (Figure 2C). It will be discussed below that the elevated open probability will favor in mitochondria a flux of K^+^ ions from the cytosol into the lumen. 

An exchange of K^+^ in the bath solution on the extracellular or the intracellular side for Ca^2+^ abolished inward and outward currents, respectively (Figure 2A,D). The current/voltage (I/V) relations exhibit in both conditions no apparent reversal voltage (Figure 2D). This underscores a high selectivity of the channel for K^+^. Hence, the permeability of the channel to Ca^2+^ is neglectable; moreover, Na^+^ is not conducted [22]. 

With these properties, the two channels are ideal tools for engineering and testing the light-controlled expression of a K^+^ channel in the mitochondria. Both channels are sorted to the mitochondria but only Kmpv_12T_ is an active channel. In this way, Kesv serves as a negative control, which reports only the impact of a closed channel in the mitochondria. The second channel, contrastingly, can be used to investigate the effect of an active and highly K^+^ selective channel in the mitochondria. In this respect, the present system differs from an expression of channelrhodopsin in mitochondria since the latter is not selecting among cations and also conducts Ca^2+^ [6,7]. 

### 3.1. Light-Induced Transcription of K^+^ Channels 

In the next step, we established the light-dependency of the transcription system. To keep the mitochondria undisturbed by the activity of the channel, we first used the inactive Kesv for these investigations. Notably, a depolarization of the inner mitochondrial membrane by an active K^+^ channel could potentially impair posttranslational import into the organelle [21]. 

To examine light-stimulated channel expression, HEK293 cells were transfected with three constructs, Kesv, CRY2 and CIB1 (Kesv/CRY2/CIB1), and then kept either in the dark or exposed to distinct protocols of blue light; the latter varied in light intensity or duration. Sixteen hours after illumination, the cells were imaged for GFP expression in the mitochondria. The representative images in Figure 3A also show that control cells, which were kept in the dark, exhibit at maximal amplification of the detection system, some GFP fluorescence. Cells which were treated at *t* = 0 for 60 min with an alternating light/dark protocol on the other hand generated a bright fluorescence signal from both channel proteins in the mitochondria (Figure 3A). The results of these experiments suggest that the system Kesv/CRY2/CIB1 is not entirely tight [20]; it seems to allow a low background expression in the dark. The light treatment on the other hand generates a very strong increase in protein expression and final accumulation in the mitochondria. A close scrutiny of the images confirms that this light-triggered expression system was not altering the sorting destination of the two channels (Figure 1D). 

To quantify the effect of light on Kesv expression and to compare data from different experiments, we defined a relative fluorescent index (RFI). The gain-factor of the confocal microscope was for each image adjusted such that the image pixels covered the entire dynamic range from the threshold of detection to saturation of the detector. The RFI was then normalized to an RFI range between 1 and 2, where the lower detection limit of the multiplier at a gain of 1000 is 1 and the brightest pixels at a gain of 500 is 2. In the images in Figure 3, the RFI value is 1.06 in the dark control and 1.57 after light-stimulation.

Using this quantification, we examined with the same type of experiments, the impact of light intensity and the time of light exposure on Kesv/CRY2/CIB1 expression. The data in Figure 3B show that blue light with intensities between 6 and 120 µmol of photons caused the same increase in the RFI value implying that the system is already saturated over the entire range of light intensities. 

In an additional experiment, cells were treated with the lowest light intensity (6 µmol) for times ranging between 30 min and 6 h. After the standard 16 h of incubation, a slight increase in the RFI value as a function of time of light exposure can be detected (Figure 3C). This indicates that the shortest light pulse is already sufficient to achieve 88% of the maximal expression. From these experiments, we conclude that the expression system is very light sensitive. The quantum flux energy of 6 µmol of blue light translates into a light intensity of 0.16 mW/mm^2^. This is in the range of daylight and an order of magnitude lower than the light, which is usually used for activating channelrhodopsin [6,7]. With this low light requirement for inducing the transcription system, phototoxic effects can be ignored. 

Considering this high light-sensitivity and the broad absorption spectrum of CRY2, we reasoned that daylight might be sufficient for stimulating the expression. To test this assumption HEK293 cells transfected with Kesv/CRY2/CIB1 were kept for 10 min in ambient daylight or for 10 min under 6 µmol blue light. Imaging of the respective cells 4 h after light-treatment shows that daylight is as efficient as 6 µmol of blue light for triggering the expression of the channel (Figure 3D). This implies that the expression system is very light sensitive, which is favorable in combination with the broad absorption spectrum up to ca. 500 nm light-induced expression of proteins also deeper in the tissue.

Subsequent experiments addressed the question on how fast a light-triggered transcription system would cause the generation of channels in the mitochondria. Cells were therefore treated after transfection with Kesv/CRY2/CIB1 for 1 h with an alternating light/dark protocol and imaged before and after light-stimulation. In the experiments reported so far, cells were transfected with three individual vectors, namely for CRY2, CB1 and the channel of interest. To make this step easier, the three vectors were combined into one ([Kesv/CRY2/CIB1]) (Appendix A). This unified vector has a size of 4897 bp (4144 bp without GFP), which is still within the range of inserts for adeno associated virus (AAV) vectors [29]. 

In the experiments reported in Figure 4A, both vector systems were used to monitor light-triggered expression of the Kesv protein. Both transfections with three independent vectors Kesv/CRY2/CIB1 as well as with the unified vector [Kesv/CRY2/CIB1] were able to trigger a rapid elevation of the RFI value above the dark control. A comparison of both systems suggests that a transfection with the individual vectors is faster and already generates 0.5 h after light stimulation an appreciable increase in GFP signal above the dark background. Cells transfected with the unified vector reach the same maximal stimulation as those transfected with three individual vectors. But the former requires a longer time for stimulating channel expression. Cells transfected with the unified vector require ≥1 h to generate a GFP signal above the dark background. Interesting to note is that the elevated GFP fluorescence again disappears in both experimental settings ca. 10 h after the stimulation of transcription (Figure 4A). This implies a turnover of the protein with a life span of several h. This is interesting in the context of an application of the system for optogenetic manipulations of the mitochondria as it is essentially reversible in long-lasting experiments. 

The finding that the expression of the channels can be triggered by daylight was raising the question on the light-specificity of the expression system. From the absorption spectrum of CRY2 [30], we anticipate that a red light of 660 nm should not trigger expression of the channel proteins. To test this prediction, Kmpv_12T_ was expressed with CRY2/CIB1 from the unified vector [Kmpv_12T_/CRY2/CIB1] in HEK293 cells. Cells were incubated for 16 and 4 h prior to imaging and exposed to single light pulses (120 µmol) of blue light (from 5 s to 120 s) or for 1 min of red light. It occurs that the shortest blue light pulse is already sufficient for a maximal activation of channel synthesis (Figure 4B). A more than 10 times longer exposure to red light on the other hand had no impact on channel expression. The fluorescence value of the red-light-treated cells is not different from the dark control. The results of these experiments underpin the assumption that the expression of the channels is specifically determined by light, which is absorbed by CRY2. 

### 3.2. Localization of Kesv and Kmpv_12T_ in Inner Mitochondria Membrane

To test whether the channel is indeed sorted and incorporated into the inner mitochondrial membrane, we employed a multistep-localization assay [6,31]. The images in Figure 5A illustrate HEK293 cells transfected with [Kesv/CRY2/CIB1]. The images show that the GFP tag on the Kesv channel remains unaffected by treating cells before and after permeabilization with digitonin. Moreover, the addition of trypan blue to permeabilized cells has no impact on the GFP signal. Collectively the results of these experiments confirm that the GFP-tagged channel is neither in the cytosol nor in the outer membrane of the mitochondria with GFP facing towards the cytosol [31].

The second assay was designed to detect a sorting of the channel to the inner membrane of the mitochondria. The series of experimental manipulations in Figure 5B underpin that only trypan blue is able to quench the GFP fluorescence after permeabilization of all membranes including that of the inner mitochondria membrane. The same results were obtained with HEK293 cells expressing Kmpv_12T_. Moreover, in this case, trypan blue only caused major bleaching of the GFP fluorescence after permeabilization of all membranes, including that of the inner mitochondria membrane (Figure 5C). This effect of trypan blue can only be explained by a location of the GFP-tagged Kesv and Kvmp_12T_ channels in the iMM with the fluorophore facing the inner lumen [31].

With the information from these experiments, we can predict that Kmpv_12T_ functions in the iMM as an outward rectifier. Since the small viral channels have a high propensity for inserting with the c- and *n*-terminal ends into the bilayer [25,32], these termini are facing in the experiments of Figure 2 the trans side of the bilayer. Following the sign conventions for transport across endomembranes, in which the lumen of the mitochondria corresponds to the cis side [33], the channel will function in vivo because of its distinct voltage dependency as an outward rectifier. This means that the voltage-dependent opening at negative voltages (Figure 2) will favor the flux of K^+^ ions from the cytosol into the mitochondria. This in turn will dissipate the energy stored as a protomotive force [8].

### 3.3. Effect of K^+^ Channels on Mitochondria Physiology

To examine the differential impact of an active versus an inactive K^+^ channel in the inner membrane of mitochondria, we monitored the morphology of the organelles in Cos7 cells, which express the respective channels. Cos7 cells were chosen for this study because their flat geometry and the size of their mitochondria favors the examination of morphological differences. 

Cos7 cells transfected with either Kesv or Kmpv_12T_ exhibit the same mitochondrial sorting pattern as in HEK293 cells (Figure 6). Kesv shows the best sorting to the mitochondria with a very low background fluorescence in the cytosol compared to cells expressing Kmpv_12T_. 

At a higher magnification, it is visible that the channels have different impacts on the mitochondrial morphology (Figure 6). While cells with the non-active Kesv channel contain mitochondria with a long, linear and branched geometry, the cells with the active channel Kmpv_12T_ show short and spherical mitochondria. The results of these experiments further confirm that Kesv is a non-active channel, whereas Kmpv_12T_ is an active channel. This activity presumably affects mitochondrial metabolism and alters the morphology of the organelles. Long and branching mitochondria found in cells expressing Kesv, are typical for healthy and non-stressed cells with balanced fusion and fission activity [34]. The fragmented phenotype evoked by Kmpv_12T_ on the other hand is associated with mitochondrial depolarization or uncoupling [35]. 

After establishing a robust system for a light-induced expression of K^+^ channel proteins in the iMM, we examined their impact on the physiological parameters of these organelles. Based on previous indirect data, we assume that an active K^+^ channel in the iMM should depolarize this membrane [8], while the inactive one should not. To test this prediction, cells were incubated with MitoTracker™ Red CMXRos, a voltage-sensitive dye, which reports the membrane voltage of mitochondria [36]. The characteristic images in Figure 7A show that the mitochondria of HEK293 cells, which are not transfected, exhibit a bright fluorescent signal from CMXRos indicating a negative voltage of the inner membrane of the mitochondria. The uncoupler CCCP quenches the fluorescence by depolarizing the mitochondria (Appendix A). In cells transfected with [Kesv/CRY2/CIB1], the CMXRos fluorescence of the mitochondria is not different from that of the surrounding un-transfected controls (Figure 7A,B); the ratio of red fluorescence from cells expressing the Kesv and un-transfected cells is close to 1. This underpins the aforementioned assumption that this channel is inactive (Appendix A) and that the insertion of a protein does not per se affect the charge status of the mitochondria. The situation is different in cells transfected with [Kmpv_12T_/CRY2/CIB1]. In the latter cells, the CMXRos fluorescence is much lower than that of the surrounding controls (Figure 7C,D). The ratiometric value of <1 means that the expression of an exogenous active K^+^ channel in the mitochondria causes a depolarization. The present data are not allowing a quantification of the amplitude of depolarization. However, the drop in fluorescent intensity in the mitochondria expressing the channel is in the same order of magnitude to that evoked by CCCP (Figure 7B). This suggests that a high K^+^ channel activity in the iMM causes a stronger depolarization than the 3–14 mV that are obtained by activators of endogenous mitochondria K^+^ channels [10]. This large depolarizing effect of Kmpv_12T_ implies that the additional exogenous K^+^ conductance exceeds that of the endogenous K^+^ channels [8,9,10,11]. At this point, it is not possible to fully explain this phenomenon. One reason for the depolarizing effect of Kmpv_12T_ could be that the exogenous channels are present at a higher density in the iMM than the endogenous channels. Another reason could be related to the relatively high unitary conductance and open probability of Kmpv_12T_, which may exceed that of most endogenous channels. This explanation is interesting in the context of the fact that many of the known endogenous K^+^ channels are gated by distinct physical and chemical factors such as voltage, mechanical stress and ligands [8,9,10,11] with the effect that they may be mostly closed under normal resting conditions such as those used in our assay. 

There is no difference in the ability of Kmpv_12T_ for inducing depolarization between experiments in which the channel was expressed in a constitutive manner or under the control of light [Kmpv_12T_/CRY2/CIB1] (Figure 7B). In the latter case, cells were incubated for 16 h and illuminated with a single 1-min light pulse of 6 µmol 4 h prior to imaging. 

Scrutiny of cells in which the channel was accumulating in the mitochondria on a large background of non-sorted protein (Figure 1C) also shows that the mitochondria in these cells are depolarized (Appendix A). This suggests that the channel protein is also in these cells entering the mitochondria and that the remaining missorted protein is not interfering in a negative manner with the activity of the channel in the mitochondria.

With a similar assay, we also examined the effect of an active K^+^ channel in iMM on the Ca^2+^ concentration in the organelles. Since Kmpv_12T_ has no Ca^2+^ permeability (Figure 2), these experiments will show the impact of voltage on the content of Ca^2+^ in the mitochondria. It is worth noting that this is different from experiments with a mitochondrial-localized channelrhodopsin. The latter also depolarizes the iMM but they are also able to conduct Ca^2+^ [17]. For our experiments, HEK293 cells were transfected either with [Kesv/CRY2/CIB1] or [Kmpv_12T_/CRY2/CIB1] and stained with Rhod-2 AM, a dye that reports the Ca^2+^ concentration of the mitochondria [37]. The representative images in Figure 7C show that the Rhod-2 signal is high in un-transfected control cells, suggesting a high Ca^2+^ concentration. This level of Rhod-2-fluorescence remains unaffected by the expression of the inactive Kesv channel, again indicating that this inactive channel has no impact on the physiology of the mitochondria (Figure 7C,D). When, on the other hand, cells express constitutively or under the control of light, the active Kmpv_12T_ channel, the fluorescence in the mitochondria decreases significantly (Figure 7C,D). The results of these experiments imply that a depolarization of the mitochondrial voltage by an activation of a highly K^+^ selective channel causes a decrease in the Ca^2+^ concentration. Collectively, the results are in good agreement with the assumption that Ca^2+^ uptake into mitochondria is driven by the voltage across the iMM [8,9]. Distinct activation and inactivation of K^+^ channels in this membrane can therefore modulate the Ca^2+^ concentration in the matrix. Such a mechanism could be of physiological importance in the context of the regulatory role, which is played by the mitochondrial Ca^2+^ homeostasis in cells [38]. One plausible scenario is that an activation of mitochondrial K^+^ channels is cytoprotective because of its impact on the mitochondrial Ca^2+^ load. This may be in particular important in the protection against ischemic injury in cardio-tissue, which is attenuated by prevention of mitochondrial Ca^2+^ overload [8]. 

Experiments in Figure 3 and Figure 4 indicated a small leakiness of the light-controlled transcription system with the effect that some channels may be synthesized also in the dark. To determine the impact of this unspecific dark effect on the physiological conditions of the mitochondria experiments of Figure 7A,C were repeated with cells kept in the dark. An analysis of the voltage- and Ca^2+^-sensitive dyes in cells, which exhibited under these conditions a low but visible GFP signal (Appendix A), shows that their fluorescent intensity is similar and even slightly higher than that of the un-transfected neighbors (Figure 7B,D). The data do also not reveal any appreciable difference between experiments with the active versus the inactive channel. Hence, any potential leakiness of the system in the dark is irrelevant for the physiological status of the mitochondria.

Previous work in which the light-activated channelrhodopsin was expressed in the iMM has shown that activation of this non-selective cation channel triggers apoptosis [7]. Here, we want to test if the activation of a K^+^ selective channel and the consequent depolarization of the iMM, which is more similar to the endogenous mitochondrial K^+^ channels, is sufficient to trigger apoptosis. The [Kmpv_12T_/CRY2/CIB1] construct was therefore expressed in HEK293 cells under control of light. After the green GFP signal indicated a successful expression, we tested apoptosis in cells with AnnexinV staining [39]. The representative image in Figure 8A shows two cells expressing [Kmpv_12T_/CRY2/CIB1]. These cells are not AnnexinV positive, while another cell, which is not expressing the channel, is stained by the marker. The results of these experiments underpin that the expression of the active K^+^ channel is not causing per se apoptosis. This conclusion is supported by a differential examination of a large number of cells transfected with either [Kmpv_12T_/CRY2/CIB1] or [Kesv/CRY2/CIB1]. The data confirm that expression of the active Kmpv_12T_ channel does not increase the probability of finding the same cell AnnexinV positive (Figure 8B). The results of these experiments underscore that an activation of a K^+^ channel and the following depolarization is not a trigger for apoptosis. The reduced Ca^2+^ load in depolarized mitochondria could even have an antiapoptotic effect in that this condition prevents activation of the permeability transition pore and hence apoptosis [10].

## 4. Conclusions

It has long been known that the iMM in many organisms from protists to humans contains a diverse set of K^+^ channels [8,9,10,11,12]. While many studies have established a cytoprotective function of these channels, the causal relationship between K^+^ channel activity and downstream events, which lead to cytoprotection, is not yet fully understood. One reason for this is the limited number of experimental tools, which are available for a control of mitochondrial K^+^ channel activity. Frequently used pharmacological openers of endogenous mitochondrial K^+^ channels have only small effects and generate in addition non-specific side effects which confound the interpretation of experimental data [8]. In two previous studies, channelrhodopsisns (mChR2) were engineered for targeting them to the iMM. This allowed precise temporal and spatial control over the permeability of the iMM. Pilot experiments showed that a light-triggered activation of mChR2 evoked not only a depolarization of the iMM but elicited a scope of physiological reactions ranging from ATP synthesis to the induction of apoptosis [6,7]. While these experiments allowed for elegant control over the ionic conductance of the iMM, they provide limited information on the role of mitochondrial K^+^ channels because the latter are, unlike channelrhodopsin, K^+^ selective and prevent diffusion of other cations such as Na^+^ and Ca^2+^. This means that neither the amplitude of the depolarization nor the ion fluxes in an activated mChR2 will be the same as in the case of an active K^+^ channel. Here, we present a versatile optogenetic platform, which allows for a light-triggered increase in the K^+^ conductance of the iMM. The system is based on a reversible light-inducible transcription of a K^+^ channel, which is eventually sorted to the iMM. The channel is detectable already less than 1 h after stimulation with blue light of very low intensity and disappears ca. 10 h later. Proof of concept experiments confirm that activity of Kmpv_12T_ in the iMM is able to affect the physiology of the mitochondria in that it depolarizes the organelles, lowers the Ca^2+^ content and changes the morphology of the mitochondria. Unlike in the case of mChR2, channel activity is not achieved directly by the light-sensitive gating of the protein but by a rather slow light-triggered transcription control system. This means that the activity of the K^+^ channel is, unlike in the case of mChR2, not responding in the time frame of ms/s but rather in the time window of one h. This means that the system is not suitable for an acute modulation of channel activity. However, since physiological conditions such as ischemia, starvation or other forms of stress, in which mitochondrial K^+^ channels may be activated, are longer lasting events, a high temporal resolution may not be essential.

## Figures and Tables

**Figure 1 cells-09-02507-f001:**
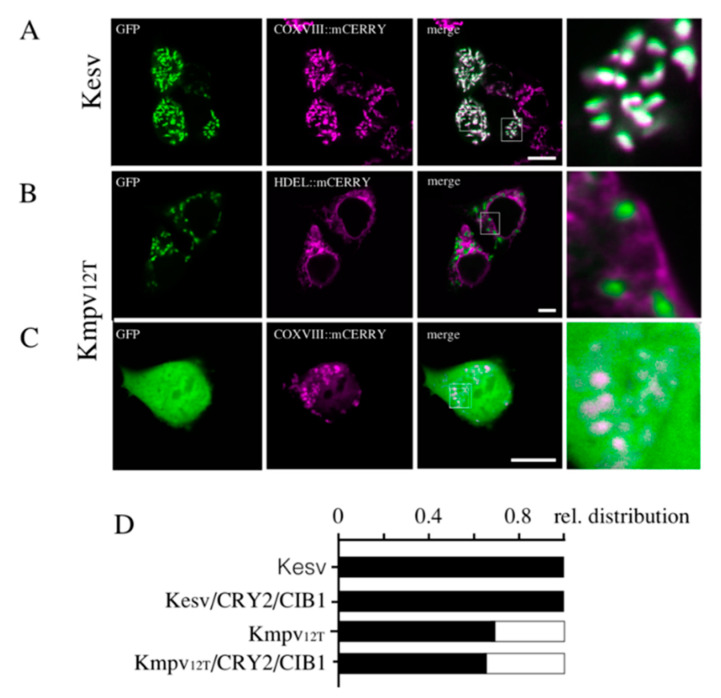
Kesv and Kmpv_12T_ are sorted to mitochondria. Fluorescent images of HEK293 cells transfected with GFP-tagged Kesv (**A**) or Kmpv_12T_ (**B**,**C**). Images show: GFP fluorescence (green, first column), fluorescence from mitochondrial marker (purple, COXVIII::mCherry) or endoplasmic reticulum (ER) marker (purple, HDEL::mCherry) (second column) and merge of purple and green channels (third column). Magnification from areas marked in merged images is shown in the fourth column. Scale bars: 10 µm. (**D**) Relative distribution of HEK293 cells in which the GFP fluorescence was only present in mitochondria as in (black bars) (**A**,**B**) or cells in which fluorescence was detected throughout (open bars) as in (**C**). The channel proteins were expressed constitutively or after triggering expression by light (Cry2/CIB1). Kesv (N = 4, *n* = 150 cells), Kmpv_12T_ (N = 2, *n* = 100 cells) Kesv/CRY2/CIB1 (N = 22, *n* = 440 cells), Kmpv_12T_/CRY2/CIB1 Kmpv_12T_ (N = 11, *n* = 228 cells).

**Figure 2 cells-09-02507-f002:**
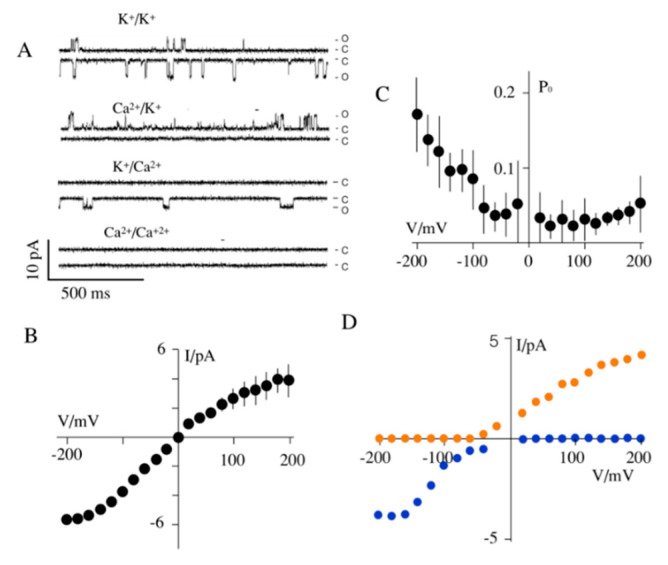
Kmpv_12T_ is a functional K^+^ channel but Kesv is not. (**A**) Exemplary single channel fluctuations of Kmpv_12T_ in a phosphatidylcholine bilayer. Channel fluctuations between the closed (c) and open (o) level at ± 120 mV were recorded in solution with symmetrical 100 mM KCl (K^+^/K^+^), symmetrical 100 mM CaCl_2_ (Ca^2+^/Ca^2+^) as well as with 100 mM KCl/CaCl_2_ on cis/trans (Ca^2+^/K^+^) or trans/cis (K^+^/Ca^2+^) side. Mean single channel current/voltage (I/V) relations (**B**) and open probability/voltage relation (P_0_/V) (**C**) from measurement in symmetrical 100 mM K^+^. (**D**) I/V relation from measurements with 100 mM KCl/CaCl_2_ on cis/trans (blue) or trans/cis (orange).

**Figure 3 cells-09-02507-f003:**
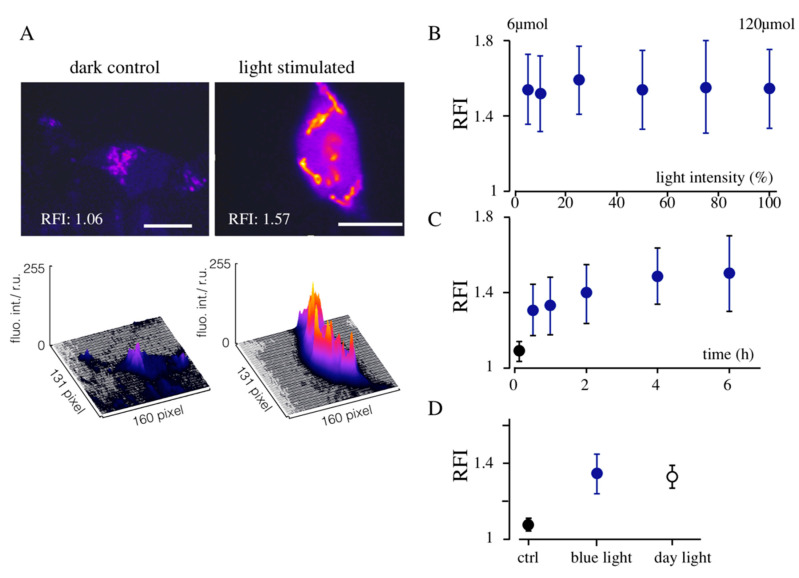
Light-triggered expression of Kesv. (**A**) Typical images of GFP fluorescence in HEK293 cells transfected with Kesv/CRY2/CIB2. Images from cells kept in the dark (left column) and cells exposed for 1 h to 6 µmol blue light (right column). To better compare the weak fluorescence of the control and the high fluorescence of illuminated cells, GFP fluorescence is presented in false color images. The reconstructed 3D surface plots in the second row report the corresponding fluorescent intensities. Relative quantification of the fluorescence (see the materials and methods) gives a relative fluorescence intensity (RFI) value of 1.06 and 1.57 for the dark control and the light-exposed cells, respectively. Scale bars: 10 µm. (**B**) Mean RFI values (±SD) from HEK293 cells transfected with Kesv/CRY2/CIB1 system. Cells were illuminated after transfection for 16 h with pulsed blue light (10 s on/60 s off) with light intensities ranging from 6 µmol (5%) to 120 µmol (100%). (**C**) Influence of the duration of illumination between 0 and 6 h on RFI value. Cells were illuminated with pulses (10 s on/60 s off) of 6 µmol blue light. Cells were incubated after transfection for a total of 16 h. Illumination was in all cases ended 4 h prior to imaging. (**D**) Mean RFI values (±SD) from HEK293 cells expressing Kmpv_12T_/CRY2/CIB1. Cells were kept in the dark (ctrl) illuminated for 10 min with 450 nm blue light (6 µmol) or kept in daylight for 10 min. In C and D, cells were incubated after transfection for a total of 16 h. Illumination was in all cases ended 4 h prior to imaging. Data are means ± standard deviation from 20 cells.

**Figure 4 cells-09-02507-f004:**
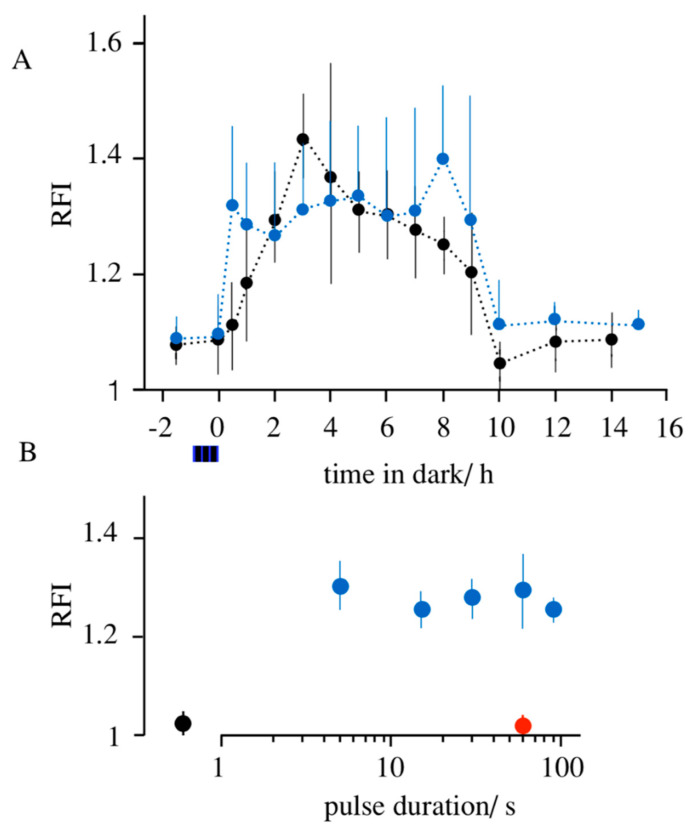
Expression of Kesv as a function of different light protocols. (**A**) Changes in RFI value before and after illuminating HEK293 cells transfected with independent vectors Kesv/CRY2/CIB1 (blue) or from a unified vector [Kesv/CRY2/CIB1] (black). Cells were illuminated for one hour with pulsed light (10 s on/60 s off, 6 µmol) before transferring them back to the dark. Cells were then imaged after variable times in the dark. The time of illumination and subsequent imaging was chosen such that all cells were incubated for a period of 16 h after transfection. Data are mean RFI values ± SD from 20 cells for each time point. (**B**) Changes in RFI value in response to single light pulses. HEK293 cells transfected with [Kesv/CRY2/CIB1] were either kept in the dark (black symbol) or illuminated with a single blue light pulse (120 µmol) between 5 and 1200 s or 60 s of red light pulse (120 µmol, 660 nm) 4 h prior to imaging. In all cases, cells were incubated for a total time of 16 h post transfection. In A and B, each data point is the mean ± SD of 20 cells imaged.

**Figure 5 cells-09-02507-f005:**
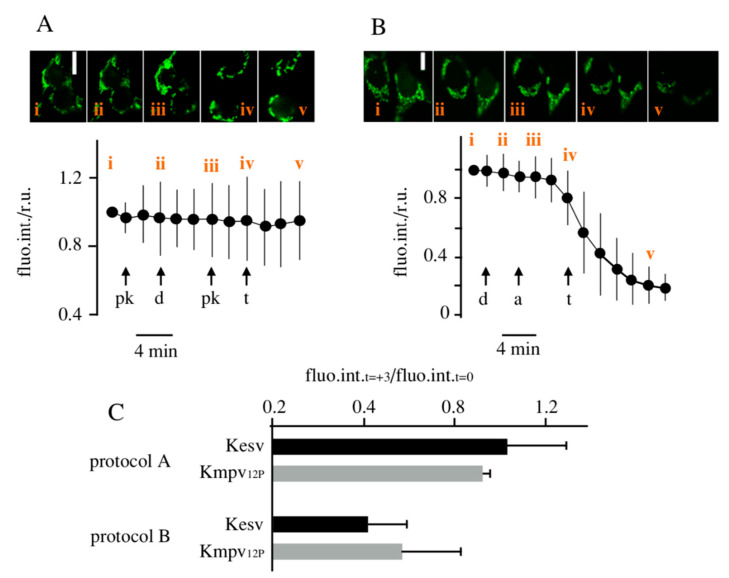
Multistep localization assay confirms the presence of Kesv and Kmpv_12T_ in inner mitochondria membrane. Cells transfected with [Kesv/CRY2/CIB1] were treated sequentially by adding protein kinase K (pk), digitonin (d), trypan blue (t), and alamethicin (a) at times indicated by arrows in (**A**,**B**). Before and after treatments, the fluorescent intensity of GFP (fluo.int.) in cells was recorded at defined intervals and normalized to value at the start of the experiment. Data points are the mean ± SD of N ≥ 4 with independent experiments with ≥21 cells. (**C**) Ratio of normalized fluorescence intensity from HEK293 cells expressing Kesv or Kmpv_12T_ measured 3 min after adding trypan blue divided by intensity immediately before treatment (fluo.int_t = +3_/fluo.int_t = 0_). Data are the means ± SD of N = 5 independent experiments with *n* > 39 cells using either protocol from A (protocol A) or B (protocol B). Scale bars: 10 µm.

**Figure 6 cells-09-02507-f006:**
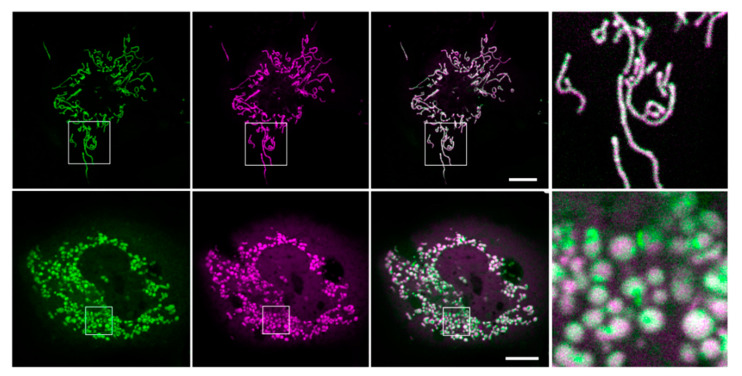
Active Kmpv_12T_ but not inactive Kesv alters the morphology of mitochondria. Confocal images of Cos7 cells expressing GFP-tagged Kesv (top) or Kmpv_12T_ (bottom). First column GFP (green), second column COXVIII::mCherry (purple), third column merge of purple and green channels. Area indicated by box is magnified in right column. Scale bars 10 µm.

**Figure 7 cells-09-02507-f007:**
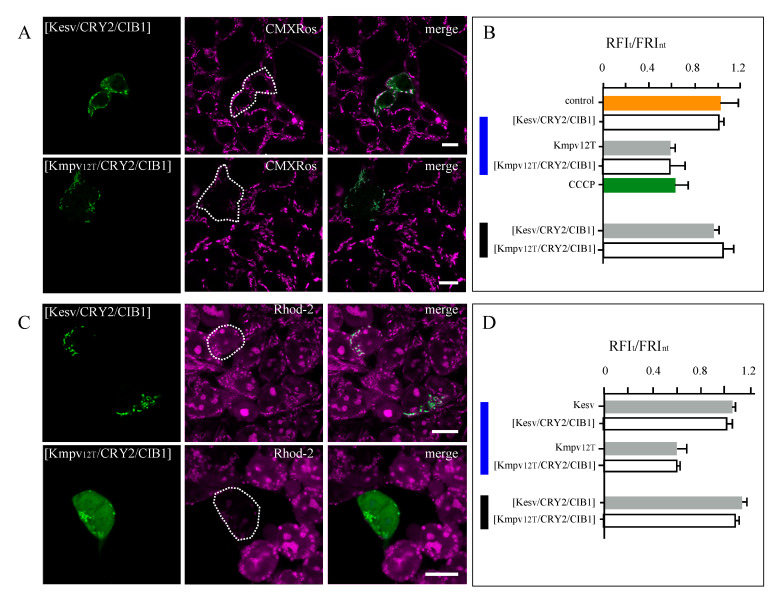
Active Kmpv_12T_ but not inactive Kesv causes the depolarization of mitochondria and a decrease in organelle Ca^2+^. Fluorescent images of HEK293 cells transfected with GFP-tagged [Kesv/CRY2/CIB1] (**A**) or [Kmpv_12T_/CRY2/CIB1] (**C**) (green, 1st column) and stained with MitoTracker™ Red CMXRos (purple, 2nd column (**A**), or Rhod-2 AM (2nd column, **C**). Overlays of GFP (green) and CMXRos or Rhod-2 signals (purple) are reported in the third column. The contours of GFP-positive cells in CMXRos and Rhod-2 channels are indicated by dashed lines. (**B**,**D**) Ratios of CMXRos or Rhod-2 signal from GFP-positive cells (RFI_t_) divided by the fluorescence of adjacent non-transfected cells (RFI_nt_). For reference, the ratio was also obtained in (**B**) from the CMXRos channel measured in untransfetced cells in the presence (green) and absence (orange) of uncoupler carbonyl cyanide m-chlorophenylhydrazone (CCCP). Data from HEK293 cells expressing Kesv or Kmpv_12T_ in constitutive manner (gray) or under control of light (open bars) with light stimulation (blue bar) or without (black bar). In the former case, cells were incubated for 16 h and illuminated with a single 1-min light pulse of 6 µmol 4 h prior to imaging (blue bar). In the latter, control experiment cells were incubated for 16 h before imaging (black bars). Data ± SD are means of N ≥ 3 independent experiments with ≥200 cells each. Scale bars 10 µm.

**Figure 8 cells-09-02507-f008:**
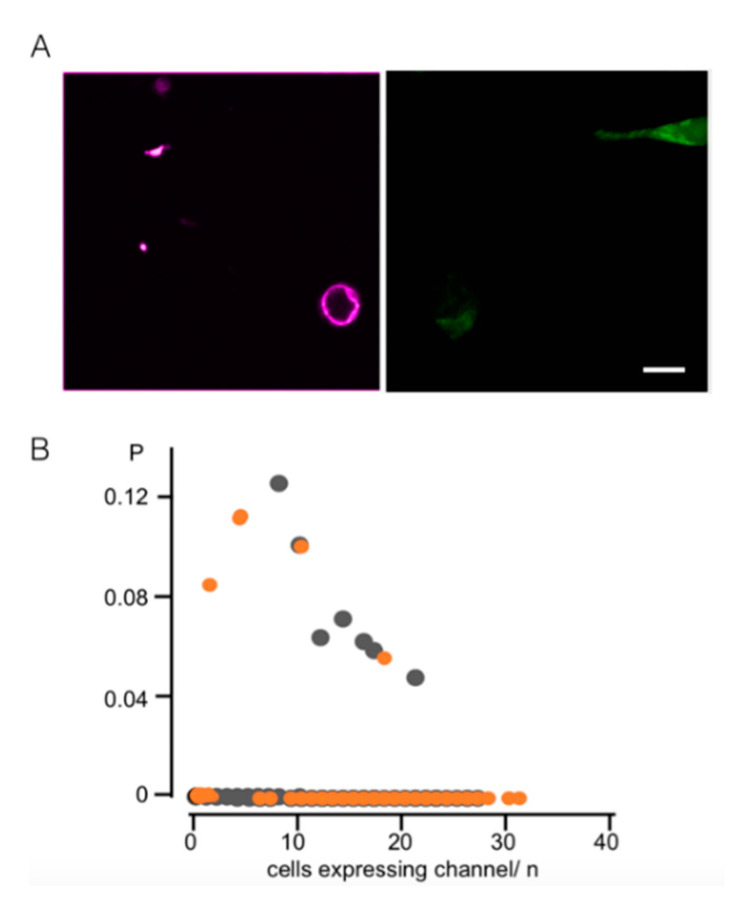
Active Kmpv_12T_ channel does not trigger apoptosis: (**A**) Fluorescent images of HEK293 cells; apoptotic cells are identified by Annexin V Alexa 457 (purple channel, left image) and cells expressing Kmpv_12T_ by GFP fluorescence (green channel, right image). The channel expressing cells are not apoptotic. Scale bar 10 µm. (**B**) Probability of finding an AnnexinV positive signal in a cell as a function of the number of HEK293 cells transfected with [Kmpv_12T_/CRY2/CIB1] (orange) or [Kesv/CRY2/CIB1] (gray) in the optical window. This analysis was performed on 400 coverslips with >4000 cells for each channel. Cells were incubated for 16 h and illuminated with a single 1-min light pulse of 6 µmol 4 h prior to imaging.

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
