# Peer review of "Light-Regulated Transcription of a Mitochondrial-Targeted K+ Channel"

_cells, 2020, doi:10.3390/cells9112507_

Round 1

Reviewer 1 Report

The paper by Engel and colleagues studies how is modulated the transcription of a light-regulated potassium channel targeted to mitochondria. The study of mitochondrial potassium channels function is very important to better understand the role fo these channels in organelle's function modulation. Furthermore, these channels have been associated with several human diseases ranging from cardiac function to cancer. The paper is well designed and written.

The only criticism is the font of the text shown in the figures that is difficult to be read. 

Author Response

Reviewer 1

The only criticism is the font of the text shown in the figures that is difficult to be read. 

Our reply:

We have changed the font in the figures hoping that the conversion into pdf is not corrupting this font.

Reviewer 2 Report

  • A brief summary (one short paragraph) outlining the aim of the paper and its main contributions.
    • This manuscript describes a proof-of-concept for a light-activated expression of a potassium selective, mitochondrial localized, cation channel. The authors clearly demonstrate a light-activated response for both Kesv and Kmpv12t that localizes to the mitochondria.  The dose of light and the duration of light application appear to have a small effect after the initial application, whereas the time following transfection does play a role.  They further show that Kmpv12t functions as a selective potassium channel and alters the morphology of the mitochondria, consistent with depolarization of the membrane.  The mitochondrial membrane depolarization is then supported by a demonstrated change in mitochondrial calcium concentration.  And finally the evidence that apoptosis is not necessarily triggered by potassium driven depolarization.

  • Broad comments highlighting areas of strength and weakness. These comments should be specific enough for authors to be able to respond.
    • Strengths: 
      • Clearly written and follows a logical progression from laying need for investigating mitochondrial membrane ion channels, to the development of a light inducible tool, to the finding that mitochondrial membrane depolarization by potassium does not necessarily lead to apoptosis.
      • Introduces a new optogenetic system that could have broad uses in several other areas of investigation.
      • Provides new insights into the depolarization of the mitochondria by potassium flow.
    • Criticisms:
      • In the manuscript as written, it states that the functional channel (Kmpv12t) is located on the inner mitochondrial membrane, but the text of section 3.2 and the figure 5 title indicate that it is the non-functional channel (Kesv).  Several conclusions are based on the notion that Kmpv12t is in the inner membrane, but that is not evident from the manuscript.
        • Sections 3.2 & Figure 5 are mixed in their use of Kesv and Kmpv12t.  The section title,  figure title, and the first two paragraphs (lines 250-263) are about Kesv, whereas the figure legend and the third paragraph (lines 264-271) are about Kmpv12t.  Was this experiment done with Kmpv12t
        • If it is a simple mix up and the experiment was done with Kmpv12t, just mislabeled in the figure title and section 3.2, I have no reservations conclusions drawn from it.  If it is not, I would like to see that additional experiment.  This needs clarified.
      • Red/Green figures are challenging for those with color vision deficiencies (CVD).  I highly recommend not using the red and green for mCherry and GFP, despite the obvious allusions to the fluorophores.   Here are some helpful items to aid in recoloring: 
        • Optimizing colormaps with consideration for color vision deficiency to enable accurate interpretation of scientific data (https://journals.plos.org/plosone/article?id=10.1371/journal.pone.0199239)
        • Coloring for colorblindness (https://davidmathlogic.com/colorblind/#%23D81B60-%231E88E5-%23FFC107-%23004D40)
      • The authors describe the light-inducible system construction in some detail in the methods section, but I think a brief discussion of the mechanism by which the system operates would be valuable to begin the results and discussion section.  This should expand on the single sentence that is in the abstract.
      • It is clear that time and effort were placed into the text of the manuscript as it has a great narrative flow, but I found the figures and figure legends lacked the same attention to detail before submission. 

  • Specific comments referring to line numbers, tables or figures.
    • Line 77:  Don’t use GOI, it’s not yet a widely accepted abbreviation.
    • Line 78:  Please include the relevant information for the K channels (Kesv & Kmpv12T), like NCBI accession numbers and virus from which they were derived.
    • Line 107:  microEinstein is not an SI unit, use micromol instead.
    • Line 373-374:  HEK293, not HKE293 cells,  put "(black bars)" before “as in A and B”. 
    • Line 379: Follow conventional order of top-to-bottom, left-to-right for A through D.
    • Line 408: Remove the "(bl)" and "(dl) light".
    • Line 411: there is an odd mark in the middle of the figure, please remove.
    • Line 429: “pk” not “pK”  Also indicate the size of the scale bar.
    • Line 439: Recolor the figure to be CVD compliant.  Also in B & D, the use of blue bars for your graph and a blue bar on the side is not a good choice.  I am also concerned that the bar second from the bottom in both B & D is incorrectly coded.  The bar name suggests it is inducible, whereas the color suggests it is constitutive expressed.

Author Response

Reviewer 2

In the manuscript as written, it states that the functional channel (Kmpv12T) is located on the inner mitochondrial membrane, but the text of section 3.2 and the figure 5 title indicate that it is the non-functional channel (Kesv).  Several conclusions are based on the notion that Kmpv12T is in the inner membrane, but that is not evident from the manuscript.

Our reply:

We had originally done the localization studies indeed only with Kesv assuming that the two similar channels behave in the same manner. However, we accept the reviewer’s criticism and we performed the same experiments also with Kmpv12T he results are the same and reported in Fig. 5C.

Reviewer 2:

Sections 3.2 & Figure 5 are mixed in their use of Kesv and Kmpv12t.  The section title,  figure title, and the first two paragraphs (lines 250-263) are about Kesv, whereas the figure legend and the third paragraph (lines 264-271) are about Kmpv12t.  Was this experiment done with Kmpv12t?  If it is a simple mix up and the experiment was done with Kmpv12t, just mislabeled in the figure title and section 3.2, I have no reservations conclusions drawn from it.  If it is not, I would like to see that additional experiment.  This needs clarified.

Our reply:

We have indeed done a mistake; in the legend it must read Kesv. This has been corrected. But as stated above we have also performed the experiments with Kmpv12T now.

Reviewer 2:

Red/Green figures are challenging for those with color vision deficiencies (CVD).  I highly recommend not using the red and green for mCherry and GFP, despite the obvious allusions to the fluorophores.   Here are some helpful items to aid in recoloring: 

Optimizing colormaps with consideration for color vision deficiency to enable accurate interpretation of scientific data (https://journals.plos.org/plosone/article?id=10.1371/journal.pone.0199239) Coloring for colorblindness  (https://davidmathlogic.com/colorblind/#%23D81B60-%231E88E5-%23FFC107-%23004D40)

Our reply:

We have changed the color code of the figures and checked with a simulation program that the colors are visible now for color blind readers.

Reviewer 2:

The authors describe the light-inducible system construction in some detail in the methods section, but I think a brief discussion of the mechanism by which the system operates would be valuable to begin the results and discussion section.  This should expand on the single sentence that is in the abstract.

Our reply:

We have added a short description of the light sensitive system and its components in the introduction.

Reviewer 2:

It is clear that time and effort were placed into the text of the manuscript as it has a great narrative flow, but I found the figures and figure legends lacked the same attention to detail before submission. 

Our reply:

We went through the figure legends and corrected the aforementioned mistakes. We further tried to improve the text in order to remove ambiguities. 

Specific comments referring to line numbers, tables or figures.

Reviewer 2:

Line 77:  Don’t use GOI, it’s not yet a widely accepted abbreviation.

Our reply:

We are now defining GOI as gene of interest

Reviewer 2:

Line 78:  Please include the relevant information for the K channels (Kesv & Kmpv12T), like NCBI accession numbers and virus from which they were derived.

Our reply:

The information has been included in Materials and Methods

Reviewer 2:

Line 107:  microEinstein is not an SI unit, use micromol instead.

Our reply:

we are using µmol as a unit now.

Reviewer 2:

Line 373-374:  HEK293, not HKE293 cells,  put "(black bars)" before “as in A and B”. 

Our reply:

We have done this

Reviewer 2:

Line 379: Follow conventional order of top-to-bottom, left-to-right for A through D.

Our reply:

The figure has been altered according to the suggestion

Reviewer 2:

Line 408: Remove the "(bl)" and "(dl) light".

Our reply:

We have done this

Reviewer 2:

Line 411: there is an odd mark in the middle of the figure, please remove.

Our reply:

We have removed this

Reviewer 2:

Line 429: “pk” not “pK”  Also indicate the size of the scale bar.

Our reply:

we have corrected pK and included the scale bare

Reviewer 2:

Line 439: Recolor the figure to be CVD compliant.  Also in B & D, the use of blue bars for your graph and a blue bar on the side is not a good choice.  I am also concerned that the bar second from the bottom in both B & D is incorrectly coded.  The bar name suggests it is inducible, whereas the color suggests it is constitutive expressed.

Our reply:

we have converted the colors in the figures to make them suitable for readers with Daltonism. We also changed the colors of the bars to distinguish their colors from the blue/black bars which indicate dark/light incubation of cells. The concern of the reviewer on the labeling of the respective bar is not correct. The cells are indeed transfected with the light sensitive transcription system, but the cells are kept for control in the dark. To avoid misunderstanding we have reworded the legend text.

Reviewer 3 Report

The manuscript by Engel et al. describes an optogenetic platform for light-triggered expression of K+ channels in the inner mitochondrial membrane. Overall the study is well designed and executed, has adequate controls and convincing experimental support for the claims. Demonstration of the viability of the technique requires no further experiments.

However, one important shortcoming of the manuscript is the lack of discussion of the other K+ channels known to be expressed in the IMM. K+ channels of different families with different gating mechanisms have been identified in the IMM (reviewed in Szabo and Zoratti, Physiol Rev 94: 519–608, 2014), including KATP, Ca2+-activated, voltage-gated, and two pore-domain K+ channels. The functional relevance of these has also been demonstrated, for example, by the use of channel blockers. The presence and possible interaction of these channels with the newly expressed exogenous channels via feedback mechanisms is completely neglected. These channels should be mentioned in the introduction and their presence must be considered in the interpretation of the results. It should be discussed, for example, why the addition of extra exogenous K+ channels to the IMM, which already has a baseline K+ conductance via the native channels to stabilize the membrane potential, can significantly depolarize the mitochondria.  

Minor points:

  1. The font used on the figures is very hard to read, the letter spacing is very small, letters tend to touch each other. This may be specific for the downloaded pdf, but that is what the reviewer sees. A different font should be used or the spacing increased.
  2. For measuring light intensity, the use of the SI units rather than the not precisely defined Einstein would be preferable for a wider audience.
  3. In Fig S3, black and gray curves are undistinguishable, the use of another color (green or blue) would help.

The English of the manuscript is generally very good with the exception of a few typos and an unusual word order in some sentences.

For example:

Ln 138 Colocalization with an organelle specific marker underpins an exclusively sorting of Kesv into the

Ln 145 Frequent reconstitution attempts of the Kesv protein generated in none of the recordings a detectable current.

Suggestion: generated no detectable current in any of the recordings

Ln 216 The means that the expression system is very light-sensitive

Ln 239 that the system is in long lasting experiments essentially reversible.

Suggestion: the system is essentially reversible in long lasting experiments.

Ln 283 The results of these experiments further confirm that Kesv is a non-active channel whereas both Kmpv12T are active channels.

Which is the other channel?

Ln 358 This conclusion is supported by a differential examination of a large number of cells which transfected with either [Kmpv12T/CRY2/CIB1] or [Kesv/CRY2/CIB1].

Author Response

Reviewer 3:

However, one important shortcoming of the manuscript is the lack of discussion of the other K+channels known to be expressed in the IMM. K+ channels of different families with different gating mechanisms have been identified in the IMM (reviewed in Szabo and Zoratti, Physiol Rev 94: 519–608, 2014), including KATP, Ca2+-activated, voltage-gated, and two pore-domain K+channels. The functional relevance of these has also been demonstrated, for example, by the use of channel blockers. The presence and possible interaction of these channels with the newly expressed exogenous channels via feedback mechanisms is completely neglected. These channels should be mentioned in the introduction and their presence must be considered in the interpretation of the results. It should be discussed, for example, why the addition of extra exogenous K+ channels to the IMM, which already has a baseline K+ conductance via the native channels to stabilize the membrane potential, can significantly depolarize the mitochondria.  

Our reply:

We have expanded the introduction saying that 10 channels with different conductance and gating properties were by now identified in mitochondria and that they are reviewed in recent papers including those mentioned by the reviewer. In the context of our tool engineering paper we find it not appropriate to discuss all the properties of the endogenous channels.

In the discussion we underscore that the large depolarizing effect of Kmpv12T apparently exceeds the effect of the endogenous K+ channels. We cannot explain this phenomenon but suggest that the exogenous channels may be present at a higher density than the endogenous channels. As alternative explanations we also mention that the additional channels have a relatively high unitary conductance and open probability. These may exceed unitary conductance/open probability of most endogenous channels. We also mention that many of the known endogenous K+ channels are gated by distinct physical and chemical factors such as voltage, mechanical stress and ligands. Hence, they may be mostly closed under normal resting conditions while the exogenous Kmpv12T has a robust high open probability. At this point we don’t see how we could discuss more on the relationship between endogenous and exogenous channels without getting into too much speculation.

Minor points:

Reviewer 3:

The font used on the figures is very hard to read, the letter spacing is very small, letters tend to touch each other. This may be specific for the downloaded pdf, but that is what the reviewer sees. A different font should be used or the spacing increased.

Our reply:

We have changed the font in the figures hoping that the conversion into pdf is not corrupting this font.

Reviewer 3:

For measuring light intensity, the use of the SI units rather than the not precisely defined Einstein would be preferable for a wider audience.

Our reply:

we are using µmol now.

Reviewer 3:

In Fig S3, black and gray curves are undistinguishable, the use of another color (green or blue) would help.

Our reply:

we have changed the colors.

The English of the manuscript is generally very good with the exception of a few typos and an unusual word order in some sentences.

For example:

Reviewer 3:

Ln 138 Colocalization with an organelle specific marker underpins an exclusively sorting of Kesv into the

Our reply:

We have reworded/corrected the sentence

Reviewer 3:

Ln 145 Frequent reconstitution attempts of the Kesv protein generated in none of the recordings a detectable current.

Suggestion: generated no detectable current in any of the recordings

Our reply:

We have reworded/corrected the sentence as suggested

Reviewer3:

Ln 216 The means that the expression system is very light-sensitive

Our reply:

We have reworded/corrected the sentence

Reviewer3:

Ln 239 that the system is in long lasting experiments essentially reversible.

Suggestion: the system is essentially reversible in long lasting experiments.

Our reply:

We have reworded the sentence as suggested

Reviewer 3:

Ln 283 The results of these experiments further confirm that Kesv is a non-active channel whereas both Kmpv12T are active channels. Which is the other channel?

Our reply:

We have corrected the sentence.

Reviewer 3:

Ln 358 This conclusion is supported by a differential examination of a large number of cells which transfected with either [Kmpv12T/CRY2/CIB1] or [Kesv/CRY2/CIB1].

Our reply:

We have reworded/corrected the sentence.

Round 2

Reviewer 2 Report

Well done, I think you have an exciting and nicely written manuscript.  I believe the changes have resulted in a stronger, clearer, story and the color alterations in the figures are exactly what I was hoping to see.  I do however have one final question.  In the version of the manuscript I received the entire conclusion section was deleted.  I am inclined to believe this was done in error, but if it is deliberate, I recommend adding a concluding sentence or two so it does end so abruptly.  Once again, well done.

Author Response

Thank you for looking so carefully! We have no idea at which stage of the review process we deleted the conclusion. This was obviously not wanted. We have added it back to the manuscript.